# Increased Omega-3 Fatty Acid Intake is Inversely Associated with Sarcopenic Obesity in Women but not in Men, Based on the 2014–2018 Korean National Health and Nutrition Examination Survey

**DOI:** 10.3390/jcm9123856

**Published:** 2020-11-27

**Authors:** Woojung Yang, Jae-woo Lee, Yonghwan Kim, Jong Hun Lee, Hee-Taik Kang

**Affiliations:** 1Department of Family Medicine, Chungbuk National University Hospital, Cheongju 28644, Korea; kineto@naver.com (W.Y.); shrimp0611@gmail.com (J.-w.L.); airsantajin@gmail.com (Y.K.); 2Department of Food Science and Biotechnology, Gachon University, Seongnam 13120, Korea; foodguy@gachon.ac.kr; 3Department of Family Medicine, Chungbuk National University College of Medicine, Cheongju 28644, Korea

**Keywords:** omega-3 fatty acids, sarcopenic obesity, omega-3 fatty acid ratio, sarcopenia

## Abstract

(1) Background: Omega-3 fatty acids (ω3FAs) are known to improve protein anabolism, increase the sensitivity to anabolic stimuli, decrease lipogenesis, and stimulate lipid oxidation. We aim to investigate whether ω3FAs are associated with the prevalence of sarcopenic obesity (SO). (2) Methods: Data were obtained from the 2014–2018 Korean National Health and Nutrition Examination Survey. The ratio of daily ω3FA intake to energy intake (ω3FA ratio) was categorized into four quartile groups. (3) Results: The prevalence of SO from Q1 to Q4 was 8.9%, 11.3%, 11.0%, and 9.8% respectively, in men and 17.4%, 14.0%, 13.9%, and 10.1% respectively, in women. The ω3FA ratio in individuals with and without SO were 1.0% and 0.9% in men (*p*-value = 0.271) respectively, and 0.8% and 1.0% in women (*p*-value = 0.017), respectively. Compared with Q1, odds ratios (95% confidence intervals) of Q2, Q3, and Q4 of ω3FA ratios were 1.563 (0.802–3.047), 1.246 (0.611–2.542), and 0.924 (0.458–1.864) respectively, in men and 0.663 (0.379–1.160), 0.640 (0.372–1.102), and 0.246 (0.113–0.534) respectively, in women, after fully adjusting for confounding factors. (4) Conclusions: The ω3FA ratio was significantly higher in older females without SO than in older females with SO. The ω3FA ratio was associated with the prevalence of SO in elderly females.

## 1. Introduction

The greatest epidemiological trend in Korea in the 21st century is the unprecedented growth of the rapidly aging population. In general, aging leads to a progressive decrease in muscle mass (sarcopenia) and an increase in fat mass (obesity) [1]. Korea, with the most rapidly aging population in the world, may confront a massive increase in the prevalence of sarcopenic obesity (SO). Sarcopenia and obesity in the elderly are frequently related to physical disability and visceral fat accumulation, displaying a synergistic interaction that can lead to a vicious cycle [2,3]. As a result, concurrent sarcopenia and obesity in elderly people increase all-cause mortality and lead to worse health outcomes than sarcopenia or obesity alone [4].

Omega-3 fatty acids (ω3FAs) are known to improve net muscle protein anabolism by activating the mammalian target of rapamycin/ribosomal protein kinase S6 (mTORp/70s6k) signaling pathway [5,6]. This signaling pathway increases the sensitivity of responses to anabolic stimuli such as enhanced protein intake, resistance exercise, and insulin [5,6]. On the other hand, ω3FAs are also found to downregulate lipogenesis by inhibiting differentiation of adipocytes by competing with prostacyclin (PGI2) in downstream [7,8] and to stimulate basal lipid oxidation by increasing the activity of peroxisomal acyl-CoA oxidase [9,10].

In some recent studies, dietary supplementation with ω3FAs has been shown to significantly decrease muscle mass loss and obesity [5,6]. Furthermore, it is noteworthy that adequate protein intake above the recommended daily intake (RDI) alone cannot guarantee prevention of sarcopenia [11,12,13]. These previous studies, however, have some limitations in that they have been implemented only in animals and in small sample sizes of people.

Therefore, the purpose of this study is to evaluate whether the ratio of daily ω3FA consumption to daily energy intake is associated with the prevalence of SO in elderly people in Korea based on the 2014–2018 Korean National Health and Nutrition Examination Survey (KNHANES).

## 2. Materials and Methods

### 2.1. Study Population

As a nationwide representative cross-sectional survey, the KNHANES has been administered to assess the health and nutritional status of Koreans residing in Korea by the Korea Centers for Disease Control and Prevention (KCDC) since 1998, following the National Health Promotion Act. The KNHANES collects data by staged, stratified, clustered, and systematic probability sampling based on sex, age, and geographic area using household registries to represent the entire Korean population living in Korea. To date, the KNHANES has been performed in seven phases including KNHANES phases I (1998), II (2001), III (2005), IV (2007–2009), V (2010–2012), VI (2013–2015), and VII (2016–2018). Among the phases of data, we selected data from 2018 (which is the latest data) to provide timely health statistics results. However, the range of selected data was expanded to 2014 to compensate for the decreased statistical power by exclusion criteria according to the definition of sarcopenia.

To gather information such as health status, health behavior, socioeconomic demographics, laboratory test results, and nutritional status of respondents, the survey consists of three components including a health interview, a health examination, and a nutrition survey among respondents. Health interviews and health examinations are conducted by trained personnel at mobile examination centers, while nutrition surveys are conducted by dietician visits to the homes of participants [14,15].

Individuals aged 60 years or older were included in our analysis from 2014 to 2018 [2,16]. Participants who have any cancers according to the definition of sarcopenia were excluded [16]. We also excluded participants with daily protein intakes under the Recommended Daily Allowance (RDA) for Korean elderly adults (0.91 g/kg/day) to evaluate the proper effects of ω3FAs on aging-related muscle loss [17]. In the final analysis, a total of 3815 participants (1960 men and 1855 women) were included and analyzed. Approval from an institutional review board (IRB) was not required because the survey did not deal with any sensitive information, only publicly available information. Data from the KNHANES are available for free on the KNHANES website (http://knhanes.cdc.go.kr) for academic research.

### 2.2. Definitions of Sarcopenic Obesity, Handgrip Strength, Obesity, and ω3FA Ratio

Sarcopenic obesity (SO) is a combination of low muscle mass (sarcopenia) and obesity [18]. Sarcopenia is an aging-related loss of muscle mass or decrease in muscular function without other comorbidities such as malignancies [16,19]. Because data regarding muscle mass were not available in the 2014–2018 KNHANES, we used decreased muscular function instead of muscle mass loss to diagnose sarcopenia [16,19,20]. To assess muscular function, the handgrip strength test was utilized.

Handgrip strength was measured using a digital grip dynamometer (T.K.K.5401; Takei Scientific Instruments Co., Ltd., Niigata, Japan) and defined as the maximally measured grip strength out of three tries with the dominant hand of participants [21]. Differences in handgrip strength among races have been reported in previous studies. For instance, the mean handgrip strength of Asian subjects was significantly different from that of Caucasian subjects [22]. Based on this, the Asian Working Group for Sarcopenia (AWGS) recommends using the lower 20th percentile of handgrip strength of each country’s health population as the cut-off value to identify low muscle strength for diagnosis of sarcopenia instead of adopting the European-based Europe Working Group on Sarcopenia in Older People 2 (EWGSOP2) cut-off points [16]. To date, there are no standard and nationally representative handgrip strength cut-off values for sarcopenia in Korea, although those based on single-year KNHANES data have been suggested [19]. Therefore, rather than referring to one of the existing values, we determined the cut-off point for sarcopenia in Korean subjects by analyzing the latest five years of KNHANES data in person according to AWGS recommendations to enhance the reliability of our study. The cut-off value of handgrip strength for a sarcopenia diagnosis was the lowest 20th percentile of handgrip strength of people without other comorbidities such as malignancies [16,19], aged from 19 to 80 years, in both sexes (34.5 kg in men and 20.0 kg in women) [19].

Body mass index (BMI) was calculated as body weight (kg) divided by squared height (m^2^). Obesity based on BMI was defined as equal to or above 25 kg/m^2^ [23].

The ratio of daily ω3FA intake to energy intake was categorized into four quartile groups: Q1, <0.4 (both sexes), Q2, 0.4–<0.7 (both sexes), Q3, 0.7–<1.1 (men), 0.7–<1.2 (women), and Q4, ≥1.1 (men), ≥1.2 (women). Hereinafter, we will refer to the ratio of daily ω3FA intake to energy intake as the ω3FA ratio.

### 2.3. Definitions of Other Variables

Individuals who engaged in moderate physical activity over 150 min per week or who engaged in vigorous physical activity over 75 min per week comprised the group defined as having sufficient physical activity. Men who drank more than seven alcoholic beverages and women who drank more than five alcoholic beverages more than twice a week were categorized as heavy alcohol drinkers [24]. Occupational status was classified into three groups, including (1) manual workers (clerks, service and sales workers, skilled agricultural, forestry, and fishery workers, persons who operate or assemble crafts, equipment, or machines, and elementary workers), (2) office workers (general managers, government administrators, professionals, and simple office workers), and (3) others (unemployed persons, housekeepers, and students). Educational status was divided into four groups as follows: <6 years, 6–<9 years, 9–<12 years, and ≥12 years of education. Marital status was categorized into two groups of (1) married and not separated (individuals who were married and living together with their spouse without separation) and (2) single (individuals who were unmarried, separated, divorced, or widowed). Blood pressure (BP) was measured three times in subjects by a standard mercury sphygmomanometer (Baumanometer; Baum Co., Inc., Copiague, NY, USA) and a mean value of the last two measurements was used as the final BP of subjects. Total cholesterol, glucose, and aspartate transaminase (AST) levels were measured by an enzymatic method, hexokinase ultraviolet, and International Federation of Clinical Chemistry (IFCC) techniques without pyridoxal-5-phosphate (P5P), respectively (Hitachi 7600 Automatic Analyzer, Hitachi, Ltd., Tokyo, Japan). High-sensitivity C-reactive protein (hsCRP) values were measured by immunoturbidimetry (Cobas analyzer; F. Hoffmann-La Roche Ltd., Basel, Switzerland).

### 2.4. Statistical Analysis

All data on continuous variables are presented as means ± standard errors (SEs). Data on categorical variables are presented as percentages ± SEs. All sampling and weight variables were stratified by sex. Statistical software SAS version 9.4 (SAS Institute Inc., Cary, NC, USA) was used for statistical analysis. This was to account for the intricate sampling design and to provide nationally representative prevalence estimates. Survey regressions and chi-squared (χ2) tests were used to compare sexes and quartiles of ω3FA ratios. *p*-values were calculated by multiple logistic regression analyses with weighting of the survey design (adjusted with age, body mass index (BMI), total cholesterol, systolic BP, fasting plasma glucose, AST, hsCRP, smoking status, alcohol intake, economic status, marital status, education duration, occupation, physical activity, history of diabetes mellitus, history of hypertension, and protein intake). We also estimated adjusted odds ratios (ORs) and 95% confidence intervals (CIs) by multivariate logistic regression models to investigate factors associated with the sarcopenic obesity group according to the ω3FA ratio. All statistical tests were two-tailed, and statistical significance was considered at *p*-values < 0.05.

## 3. Results

Table 1 presents population characteristics according to sex. The mean ages of men and women were 68.8 and 68.5 years, respectively. Male and female daily energy intakes were 2423.1 and 2013.3 kilocalories (kcal), respectively. Men consumed more macronutrients (carbohydrates, proteins, and fats) than women (all *p*-values < 0.001). However, in terms of the ω3FA ratio, the same amount was consumed (0.9%) in both sexes (*p*-value = 0.254).

Table 2 shows participants’ characteristics according to ω3FA ratio quartile. In men, mean age was not significantly different among the four quartile groups, while daily energy intake and hsCRP levels decreased with increasing quartiles of ω3FA ratio. Daily protein intake positively correlated with ω3FA ratio quartile. In women, individuals in higher quartiles of ω3FA ratio were younger. Daily energy intake decreased with ω3FA ratio quartiles, while protein intake increased. Among the four groups, hsCRP levels were not different.

Table 3 demonstrates fat intake according to the presence of sarcopenic obesity. Values of ω3FA ratio in women are the only significant difference between groups with sarcopenic obesity and non-sarcopenic obesity. The ω3FA ratio in individuals without sarcopenic obesity was significantly higher than in individuals with sarcopenic obesity (0.8% in sarcopenic obesity vs. 1.0% in non-sarcopenic obesity, *p*-value = 0.017).

Figure 1 presents the prevalence of sarcopenic obesity according to quartile of ω3FA ratio. There was no significant difference in sarcopenic obesity prevalence among male quartile groups, while the female prevalence was marginally different (*p*-value = 0.055). The lowest quartile of ω3FA ratio had the highest prevalence (17.4%) of sarcopenic obesity in women, while the highest quartile of ω3FA ratio showed the lowest (10.1%).

Table 4 demonstrates logistic regression models according to ω3FA ratio quartile. Compared with Q1, ORs (95% CIs) for sarcopenic obesity of Q2, Q3, and Q4 of ω3FA ratio were 1.288 (0.776–2.138), 1.298 (0.804–2.097), and 1.118 (0.701–1.782) respectively, in men and 0.807 (0.534–1.221), 0.847 (0.552–1.299), and 0.603 (0.376–0.967) respectively, in women after adjusting for age. After full adjustment, ORs (95% CIs) of Q2, Q3, and Q4 of ω3FA ratio were 1.563 (0.802–3.047), 1.246 (0.611–2.542), and 0.924 (0.458–1.864) respectively, in men, and 0.663 (0.379–1.160), 0.640 (0.372–1.102), and 0.246 (0.113–0.534) respectively, in women.

## 4. Discussion

This study finds that the ω3FA ratio was significantly higher in women without SO than in women with SO. Logistic regression models demonstrate that the ω3FA ratio was inversely associated with the prevalence of SO in elderly females (but not in males), even after adjusting for confounding variables.

To date, resistance exercise and protein supplementation have been generally accepted as the most effective lifestyle changes to prevent sarcopenia [25,26]. Thus, the two are recommended for the prevention and management of sarcopenia [27,28]. However, resistance exercise and protein supplementation have not been shown to consistently lead to significantly positive results in all the research [28]. Anabolic resistance, which is a decreased response to anabolic stimuli such as physical exercise and ingested protein, arises in the protein metabolism of human bodies as people get older [29]. Thus, we have hypothesized that an additional approach for overcoming anabolic resistance would be necessary along with exercise and protein intake to prevent and treat sarcopenia in the elderly.

In the past, many researchers hypothesized that anti-inflammatory effects of ω3FAs were mainly related to observed prevention effects in sarcopenia such as cachexia in cancers or chronic diseases [30]. Today, overcoming age-related anabolic resistance is considered to be the main mechanism of ω3FAs in the prevention of sarcopenia [31]. On the molecular level, the mTORp/70s6k signaling pathway is considered an integral point of age-related anabolic resistance [6]. The mammalian target of rapamycin (mTOR) is a sensing effector of hormone and nutrient availability that controls two key translation initiation promoters, S6 kinase (S6K) and 4E binding protein (BP), of eukaryotic initiation factor eIF4E [5]. ω3FAs make this effector more sensitive to anabolic stimuli, and thus, promote cellular mRNA translation of proteins [5,31]. Prostaglandin I2 (prostacyclin, PGI2) triggers cyclic adenosine monophosphate (cAMP) production through prostacyclin receptors at the surface of preadipocytes. cAMP activates the protein kinase A pathway that promotes the differentiation of preadipocytes into adipocytes. ω3FAs inhibit the synthesis of PGI2 at the level of cyclooxygenases (COX) by competing with omega-6 fatty acids (ω6FAs), which are precursors of PGI2, and consequently inhibit lipogenesis [32,33].

ω3FAs are also found to increase the activity of peroxisomal and mitochondrial enzymes for β-oxidation of fatty acids. ω3FAs mainly induce gene expression of acyl-CoA oxidase, which is the key enzyme of fatty acids β-oxidation in peroxisomes [10]. Concurrently, ω3FAs elevate the activity of carnitine palmitoyltransferase II, which is a shuttle of fatty acids into mitochondria for further β-oxidation. Therefore, ω3FAs accelerate overall lipid oxidation in both peroxisomes and mitochondria [34].

As mentioned above, previous studies show that ω3FAs have the potential to prevent and treat SO. However, because most studies have explored the effects of ω3FAs either for sarcopenia [6] or obesity (but not for both [35]), previous studies alone cannot directly evaluate the beneficial effects of ω3FAs for SO. By directly investigating the association between ω3FA intake and the prevalence of SO, this study differentiates its clinical significance from previous studies to date.

A noteworthy point of our findings is that there is a significant association between ω3FA ratio and prevalence of SO in older women but not in men. Although the exact mechanism remains unknown, women might have more capacity for improvement with ω3FAs compared to men [36]. Unlike men, the muscle strength of women does not usually increase to its optimal level in response to resistance exercise; thereby, most women get greater muscle function reservoir to enhance with the intake of ω3FAs [37]. The innate biologically sexual differences in the pattern of muscle mass and strength changes with age may also explain these differences. Peak muscle mass and strength of women are lower than in men [38]. Furthermore, the muscle decline in women occurs earlier due to menopause and is steeper than in men, particularly after menopause [39]. Menopause is a major risk factor for sarcopenia and SO [40]. In the present study, only elderly individuals were included, indicating the possibility that most of the women were postmenopausal. Considering the differences in changes between men and women in the pattern of muscle mass and strength with aging, supplementation of muscle-preserving nutrients, including ω3FA, may show a greater preventive effect against SO in women than in men.

Comparing the intake of ω3FA to that in Western countries such as the US and European countries, Koreans are known to consume more ω3FA. We can compare this more reliably by measuring the blood level of eicosapentaenoic acid (EPA) and docosahexaenoic acid (DHA) than by the 24 h recall method because we can exclude the potential for reporting or recall bias [40,41]. The blood EPA and DHA levels in Koreans are more than twice those of US subjects and higher than most of those in European countries, except Scandinavia [42].

Several limitations should be considered in interpreting this study. First, we defined sarcopenia not directly by measuring muscle mass using body composition examination such as dual-energy X-ray absorptiometry, but indirectly, by measuring muscle function using a handgrip strength test. Because handgrip strength is a reliable predictor of clinical outcomes of low muscle mass, handgrip strength can be used as an indicator to diagnose sarcopenia [16,19]. Second, it is difficult to conclude causality from this study because it was cross-sectionally designed without any follow-up. Third, because we use the data of dietary ω3FA intake gathered by a 24 h recall method instead of actual measurements of ω3FAs in the red blood cell membrane or plasma of subjects, we cannot completely exclude the potential for reporting or recall bias from this study.

In spite of these limitations, this study has several strengths that distinguish our research from previous studies. First, this study is derived from a population-based sample from a complex survey design, seeking to produce estimates that accurately represent the Korean population. Second, to the best of our knowledge, this is the first Korean study to evaluate the association of ω3FA intake and ω3FA ratio with the prevalence of SO using nationally representative data. Third, to minimize confounding effects of several factors, especially protein levels, we analyzed the data not only by adjusting for variables such as daily protein intake but also by using the ω3FA ratio instead of measured amounts of ω3FAs. Fourth, the definition of sarcopenia was not quantitatively measured by muscle mass but instead was functionally evaluated using handgrip strength. To predict clinical outcomes in elderly people, muscular functionality such as handgrip strength is more important than the quantity of muscle mass [20,43].

## 5. Conclusions

In conclusion, this study shows that an increased ω3FA ratio-based intake is associated with a decreased prevalence of SO in elderly females. This means that interventions to increase dietary ω3FA ratio may help with the prevention and management of SO in elderly females. However, longitudinally designed cohort studies or interventional studies are further needed to confirm causality between the ω3FA ratio and the prevalence of SO and to determine whether a diet with an increased ω3FA ratio might be an effective strategy to prevent and manage SO in specific populations.

## Figures and Tables

**Figure 1 jcm-09-03856-f001:**
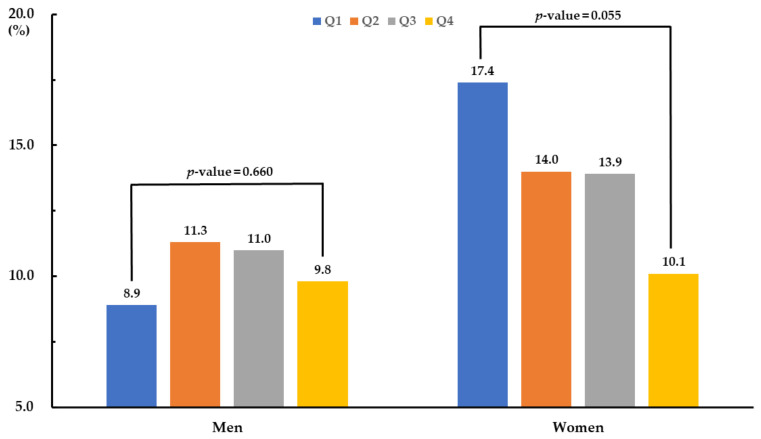
Prevalence of sarcopenic obesity according to quartile group of omega-3 fatty acid ratio.

**Table 1 jcm-09-03856-t001:** Population characteristics according to sex.

Characteristic	Males	Females	*p*-Value
Number	1960	1855	
Age, years	68.8 ± 0.2	68.5 ± 0.2	0.159
BMI, Kg/m^2^	23.5 ± 0.1	23.6 ± 0.1	0.173
Energy intake, Kcal/day	2423.1 ± 18.8	2013.3 ± 17.0	<0.001
Carbohydrate intake, g/day	381.6 ± 3.4	339.0 ± 3.4	<0.001
Protein intake, g/day	88.3 ± 0.8	73.1 ± 0.6	<0.001
Fat intake, g/day	44.6 ± 0.8	38.1 ± 0.6	<0.001
Daily omega-3/energy intake, %	0.9 ± 0.0	0.9 ± 0.0	0.254
Total cholesterol, mg/dL	182.4 ± 1.0	194.6 ± 1.1	<0.001
Systolic blood pressure, mmHg	125.0 ± 0.4	127.3 ± 0.6	<0.001
Fasting plasma glucose, mg/dL	108.6 ± 0.7	104.1 ± 0.7	<0.001
AST, IU/L	24.6 ± 0.3	23.0 ± 0.2	<0.001
hsCRP, mg/L	1.5 ± 0.1	1.3 ± 0.1	0.082
Current smoking, %	21.7 ± 1.0	2.4 ± 0.4	<0.001
Heavy alcohol intake, %	12.9 ± 0.9	1.8 ± 0.3	<0.001
Economic status, %			0.006
Low	26.6 ± 1.2	31.9 ± 1.3	
Middle–low	29.2 ± 1.2	27.8 ± 1.3	
Middle–high	23.7 ± 1.0	20.8 ± 1.1	
High	20.5 ± 1.1	19.6 ± 1.3	
Marital status, %			<0.001
Married and not separated	90.9 ± 0.7	62.2 ± 1.4	
Single	9.1 ± 0.7	37.8 ± 1.4	
Education duration, %			<0.001
<6 years	30.8 ± 1.3	52.9 ± 1.5	
6–<9 years	17.8 ± 1.1	16.8 ± 1.0	
9–<12 years	29.2 ± 1.3	18.6 ± 1.1	
≥12 years	22.2 ± 1.2	11.6 ± 1.0	
Occupation, %			<0.001
Office workers	11.4 ± 0.9	4.0 ± 0.6	
Manual workers	40.9 ± 1.4	30.5 ± 1.3	
Other	47.7 ± 1.4	65.5 ± 1.3	
Sufficient physical activity, %	46.5 ± 1.4	36.8 ± 1.4	<0.001
History of diabetes mellitus, %	17.0 ± 1.0	13.0 ± 0.9	0.002
History of hypertension, %	42.1 ± 1.3	41.5 ± 1.4	0.782

All data are presented as mean ± standard errors (SEs) or percentage ± SEs. Abbreviations: BMI, body mass index; AST, aspartate transaminase; IU/L, international units per liter; hsCRP, high-sensitivity C-reactive protein.

**Table 2 jcm-09-03856-t002:** Population characteristics according to quartiles of ω3FA ratio.

Males	Q1	Q2	Q3	Q4	*p*-Value
	(<0.4)	(0.4–<0.7)	(0.7–<1.1)	(≥1.1)	
Number	517	483	485	475	
Age, years	68.8 ± 0.3	69.4 ± 0.3	68.3 ± 0.3	68.8 ± 0.3	0.087
BMI, Kg/m^2^	23.3 ± 0.1	23.3 ± 0.1	23.7 ± 0.1	23.6 ± 0.1	0.061
Energy intake, Kcal/day	2495.8 ± 36.6	2458.4 ± 36.8	2360.2 ± 32.6	2378.4 ± 39.8	0.024
Carbohydrate intake, g/day	403.0 ± 7.3	401.7 ± 6.3	366.2 ± 5.3	355.7 ± 5.9	<0.001
Protein intake, g/day	82.7 ± 1.3	86.5 ± 1.4	88.6 ± 1.3	95.5 ± 1.9	<0.001
Fat intake, g/day	38.6 ± 1.3	42.6 ± 1.4	43.8 ± 1.3	53.2 ± 1.7	<0.001
Daily omega-3/energy intake, %	0.3 ± 0.0	0.5 ± 0.0	0.9 ± 0.0	2.0 ± 0.1	<0.001
Total cholesterol, mg/dL	185.4 ± 1.9	181.8 ± 1.8	183.0 ± 2.1	179.3 ± 2.1	0.223
Systolic blood pressure, mmHg	126.8 ± 0.8	126.2 ± 0.8	123.4 ± 0.7	123.7 ± 0.9	0.002
Fasting plasma glucose, mg/dL	111.1 ± 1.5	106.7 ± 1.3	109.1 ± 1.5	107.6 ± 1.3	0.170
AST, IU/L	25.7 ± 0.7	24.2 ± 0.4	24.3 ± 0.4	24.2 ± 0.5	0.283
hsCRP, mg/L	1.8 ± 0.2	1.5 ± 0.2	1.6 ± 0.1	1.2 ± 0.1	0.013
Current smoking, %	26.3 ± 2.3	20.0 ± 2.0	21.5 ± 2.2	19.2 ± 2.1	0.113
Heavy alcohol intake, %	16.3 ± 1.8	12.0 ± 1.6	13.0 ± 1.8	10.4 ± 1.6	0.082
Economic status, %					0.037
Low	32.2 ± 2.4	26.1 ± 2.4	22.1 ± 2.1	25.9 ± 2.3	
Middle–low	29.4 ± 2.2	29.0 ± 2.4	28.9 ± 2.4	29.4 ± 2.4	
Middle–high	19.5 ± 2.0	22.4 ± 2.1	27.2 ± 2.2	25.8 ± 2.1	
High	18.9 ± 2.1	22.5 ± 2.4	21.8 ± 2.2	18.9 ± 1.9	
Marital status, %					0.038
Married and not separated	87.7 ± 1.6	90.9 ± 1.5	93.4 ± 1.2	91.5 ± 1.5	
Single	12.3 ± 1.6	9.1 ± 1.5	6.6 ± 1.2	8.5 ± 1.5	
Education duration, %					<0.001
<6 years	39.7 ± 2.4	29.5 ± 2.3	28.7 ± 2.3	25.1 ± 2.4	
6–<9 years	17.8 ± 1.9	18.1 ± 2.0	17.3 ± 2.1	18.0 ± 2.2	
9–<12 years	25.2 ± 2.1	28.4 ± 2.4	32.3 ± 2.5	31.1 ± 2.6	
≥12 years	17.4 ± 2.1	24.0 ± 2.3	21.7 ± 2.1	25.7 ± 2.2	
Occupation, %					0.450
Office workers	10.7 ± 1.6	10.9 ± 1.7	12.7 ± 1.9	11.3 ± 1.6	
Manual workers	46.4 ± 2.6	41.3 ± 2.7	38.8 ± 2.6	37.1 ± 2.4	
Other	42.9 ± 2.6	47.7 ± 2.7	48.4 ± 2.7	51.6 ± 2.7	
Sufficient physical activity, %	44.7 ± 2.5	49.0 ± 2.8	47.3 ± 2.6	44.8 ± 2.6	0.633
History of diabetes mellitus, %	14.4 ± 1.7	15.9 ± 1.9	20.4 ± 2.2	17.3 ± 1.8	0.172
History of hypertension, %	41.8 ± 2.4	44.5 ± 2.6	39.2 ± 2.5	42.8 ± 2.7	0.523
**Females**	**Q1**	**Q2**	**Q3**	**Q4**	***p*-Value**
	(<0.4)	(0.4–<0.7)	(0.7–<1.2)	(≥1.2)	
Number	460	479	454	462	
Age, years	69.5 ± 0.4	68.9 ± 0.3	68.0 ± 0.4	67.6 ± 0.3	<0.001
BMI, Kg/m^2^	23.5 ± 0.2	23.7 ± 0.1	23.7 ± 0.1	23.7 ± 0.1	0.704
Energy intake, Kcal/day	2138.0 ± 40.5	1975.8 ± 30.0	1945.5 ± 31.7	1994.2 ± 34.4	0.002
Carbohydrate intake, g/day	388.5 ± 8.7	337.8 ± 5.8	317.4 ± 5.6	312.6 ± 6.7	<0.001
Protein intake, g/day	69.5 ± 1.5	69.4 ± 1.0	74.8 ± 1.3	78.7 ± 1.3	<0.001
Fat intake, g/day	30.1 ± 1.1	34.7 ± 1.1	40.3 ± 1.2	47.2 ± 1.4	<0.001
Daily omega-3/energy intake, %	0.3 ± 0.0	0.6 ± 0.0	0.9 ± 0.0	2.0 ± 0.1	<0.001
Total cholesterol, mg/dL	194.9 ± 2.1	195.1 ± 2.2	193.9 ± 2.0	194.4 ± 2.2	0.979
Systolic blood pressure, mmHg	129.2 ± 1.2	127.1 ± 0.8	126.1 ± 0.9	126.9 ± 1.1	0.211
Fasting plasma glucose, mg/dL	102.9 ± 1.3	106.4 ± 1.9	101.6 ± 1.1	105.5 ± 1.3	0.062
AST, IU/L	22.6 ± 0.4	23.2 ± 0.7	23.1 ± 0.4	23.1 ± 0.4	0.702
hsCRP, mg/L	1.3 ± 0.1	1.4 ± 0.2	1.4 ± 0.2	1.2 ± 0.1	0.863
Current smoking, %	4.0 ± 1.2	2.6 ± 0.9	1.2 ± 0.5	2.0 ± 0.9	0.129
Heavy alcohol intake, %	2.9 ± 0.9	1.9 ± 0.7	1.6 ± 0.7	0.8 ± 0.3	0.084
Economic status, %					<0.001
Low	36.1 ± 2.6	36.9 ± 2.5	24.6 ± 2.4	29.8 ± 2.3	
Middle–low	32.9 ± 2.5	25.9 ± 2.3	26.3 ± 2.2	26.0 ± 2.3	
Middle–high	17.7 ± 2.3	18.7 ± 2.0	26.1 ± 2.6	20.6 ± 2.2	
High	13.4 ± 2.1	18.4 ± 2.4	23.0 ± 2.2	23.7 ± 2.5	
Marital status, %					<0.001
Married and not separated	55.3 ± 2.7	56.7 ± 2.7	67.6 ± 2.6	69.2 ± 2.5	
Single	44.7 ± 2.7	43.3 ± 2.7	32.4 ± 2.6	30.8 ± 2.5	
Education duration, %					<0.001
<6 years	64.2 ± 3.1	55.0 ± 2.7	45.9 ± 2.6	47.2 ± 2.8	
6–<9 years	15.2 ± 2.1	16.3 ± 1.9	17.7 ± 1.9	17.9 ± 2.4	
9–<12 years	13.8 ± 2.3	18.8 ± 2.2	22.0 ± 2.3	19.5 ± 2.0	
≥12 years	6.8 ± 1.6	9.8 ± 1.5	14.3 ± 1.9	15.3 ± 1.8	
Occupation, %					0.782
Office workers	2.9 ± 1.1	2.9 ± 0.9	5.1 ± 1.2	4.9 ± 1.0	
Manual workers	31.8 ± 2.5	34.7 ± 2.3	26.3 ± 2.3	29.3 ± 2.2	
Other	65.4 ± 2.5	62.4 ± 2.4	68.5 ± 2.4	65.9 ± 2.3	
Sufficient physical activity, %	34.6 ± 2.7	35.5 ± 2.4	39.9 ± 2.7	36.9 ± 2.6	0.547
History of diabetes mellitus, %	13.0 ± 1.9	13.5 ± 1.9	10.5 ± 1.5	14.9 ± 1.7	0.327
History of hypertension, %	41.1 ± 2.6	41.3 ± 2.5	40.1 ± 2.6	43.6 ± 2.6	0.804

All data are presented as mean ± SEs or percentage ± SEs. Abbreviations: ω3FA, omega-3 fatty acid; ω3FA ratio, ratio of daily omega-3 fatty acid intake to energy intake; BMI, body mass index; AST, aspartate transaminase; hsCRP, high-sensitivity C-reactive protein.

**Table 3 jcm-09-03856-t003:** Ratio of daily total fat and fatty acids intake to energy intake according to the presence of sarcopenic obesity.

Ratio of Daily Total Fat and Fatty Acids Intake to Energy Intake (%)	Sarcopenic Obesity	Non-Sarcopenic Obesity	*p*-Value
Males			
Total fat	16.5 ± 0.6	16.3 ± 0.2	0.741
SFA	4.7 ± 0.2	4.7 ± 0.1	0.853
MUFA	5.0 ± 0.2	5.0 ± 0.1	0.922
PUFA	4.9 ± 0.2	4.6 ± 0.1	0.177
Omega-3 FA	1.0 ± 0.1	0.9 ± 0.0	0.271
Omega-6 FA	3.9 ± 0.2	3.7 ± 0.1	0.270
Females			
Total fat	16.8 ± 0.6	17.1 ± 0.2	0.633
SFA	4.9 ± 0.2	4.9 ± 0.1	0.879
MUFA	5.2 ± 0.2	5.2 ± 0.1	0.995
PUFA	4.6 ± 0.2	5.0 ± 0.1	0.121
Omega-3 FA	0.8 ± 0.0	1.0 ± 0.0	0.017
Omega-6 FA	3.8 ± 0.2	4.0 ± 0.1	0.334

All data are presented as mean (± standard errors). Abbreviations: SFA, saturated fatty acids; MUFA, monounsaturated fatty acids; PUFA, polyunsaturated fatty acids; FA, fatty acids.

**Table 4 jcm-09-03856-t004:** Adjusted odds ratios (ORs) and 95% confidence intervals (CIs) of sarcopenic obesity by quartiles of ω3FA ratio.

Males	Q1	Q2	Q3	Q4
	(<0.4)	(0.4–<0.7)	(0.7–<1.1)	(≥1.1)
Model 1	1 (ref)	1.288 (0.776–2.138)	1.298 (0.804–2.097)	1.118 (0.701–1.782)
Model 2	1 (ref)	1.668 (0.868–3.205)	1.160 (0.592–2.272)	1.000 (0.512–1.953)
Model 3	1 (ref)	1.563 (0.802–3.047)	1.246 (0.611–2.542)	0.924 (0.458–1.864)
**Females**	**Q1**	**Q2**	**Q3**	**Q4**
	(<0.4)	(0.4–<0.7)	(0.7–<1.2)	(≥1.2)
Model 1	1 (ref)	0.807 (0.534–1.221)	0.847 (0.552–1.299)	0.603 (0.376–0.967)
Model 2	1 (ref)	0.604 (0.355–1.028)	0.649 (0.394–1.072)	0.282 (0.129–0.617)
Model 3	1 (ref)	0.663 (0.379–1.160)	0.640 (0.372–1.102)	0.246 (0.113–0.534)

Odds ratios (ORs) and 95% confidence intervals (CIs) were calculated using weighted multivariate logistic regression analyses. Model 1: Adjusted for age. Model 2: Adjusted for age, BMI, total cholesterol, systolic blood pressure, fasting plasma glucose, AST, and hsCRP. Model 3: Adjusted for current smoking status, heavy alcohol intake, economic status, marital status, education duration, occupation, sufficient physical activity, history of diabetes mellitus, history of hypertension, and protein intake in addition to variables of Model 2. Abbreviation: ω3FA ratio, ratio of daily ω3FA intake to energy intake; AST, aspartate transaminase; hsCRP, high-sensitivity C-reactive protein.

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
