# Peer review of "Increased Omega-3 Fatty Acid Intake is Inversely Associated with Sarcopenic Obesity in Women but not in Men, Based on the 2014–2018 Korean National Health and Nutrition Examination Survey"

_jcm, 2020, doi:10.3390/jcm9123856_

Round 1
Reviewer 1 Report
The authors presented data from a large sample of older individuals and examined whether the ratio of daily ω3FA consumption to daily energy intake is associated with the prevalence of SO.
The study is well presented with a well-developed methodology according to the proposed objectives. However, there are two main concerns about the results founded. Is there a biological basis that explains the phenomenon that you have found? In other words, why does ω3FA maintain muscle mass in women but not in men?
Also, I miss the following key points to be able to discuss the findings:
TITLE:
1. If the phenomenon found only occurs in women, it would be advisable to change the title to (... "in women but no men" ...)
INTRODUCTION:
1. If the phenomenon was only found in women, it would be advisable to present evidence on why ω3FA can help maintain muscle mass in women more than in men.
METHOD:
1. Why is the Handgrip cut-off points not used as a method to classify the OS according to the latest EWGSOP2 updates?
RESULTS:
1. Why are the sarcopenia frequencies not presented according to the EWGSOP by quartiles of ω3FA?
DISCUSSION:
1. This is the most critical aspect: according to their results, the phenomenon only occurs in women. It could be argued-discussed with biological bases as to why it only happens in women but not in men? Is there evidence of this fact?
Although the article is well presented and is part of an extensive database such as KNHANES, I believe that the statistical findings are merely findings.
Reviewer 2 Report
Please discuss the potential reason of the difference between men and women. Please discuss the Impact of the economic Status on Omega-3-intake and obesity. Low Omega-3-intake is associated with obesity and social Status/education. What is the henn, what is the egg. Martial Status is associated with Omega-3-ratio strongly in females but not in males. Married females are cooking healthier Food?
Please compare the Omega-3-fatty Acid intake in Korea to the intake in Europe and US
